



# Facility for generation of ambient-like model aerosols in the laboratory: application in the intercomparison of automated PM monitors with the reference gravimetric method

Stefan Horender[1], Kevin Auderset[1], Paul Quincey[2], Stefan Seeger[3], Søren Nielsen Skov[4], Kai Dirscherl[5], Thomas O. M. Smith[2], Katie Williams[2], Camille C. Aegerter[1], Daniel M. Kalbermatter[1], François Gaie-Levrel[6] and Konstantina Vasilatou[1]

[1]Federal Institute of Metrology METAS, Bern-Wabern, 3003, Switzerland
[2]National Physical Laboratory (NPL), Teddington, London, UK
[3]Bundesanstalt für Materialforschung und -prüfung (BAM), Berlin, Germany
[4]Bioengineering and Environmental Technology, Danish Technological Institute (DTI), Aarhus, Denmark
[5]Danish National Metrology Institute (DFM), Kogle Alle 5, 2970 Hørsholm, Denmark
[6]Laboratoire national de métrologie et d'essais (LNE), Paris, France

*Correspondence to*: Konstantina Vasilatou (konstantina.vasilatou@metas.ch)

**Abstract.** A new facility has been developed which allows for a stable and reproducible generation of ambient-like aerosols in the laboratory. The setup consists of multiple aerosol generators, a custom-made flow tube homogeniser, isokinetic sampling probes and a system to control aerosol temperature and humidity. Model aerosols containing elemental carbon, secondary organic matter from the photo-oxidation of α-pinene, inorganic salts such as ammonium sulphate and ammonium nitrate, mineral dust particles and water were generated at different environmental conditions and different number and mass concentrations. The aerosol physical and chemical properties were characterised with an array of experimental methods, including scanning mobility particle sizing, ion chromatography, total reflection X-ray fluorescence spectroscopy, and thermo-optical analysis. The facility is very versatile and can find applications in the calibration and performance characterisation of aerosol instruments monitoring ambient air. In this study, we performed, as proof of concept, an intercomparison of three different commercial PM (particulate matter) monitors (TEOM 1405, DustTrak DRX 8533 and Fidas Frog) with the gravimetric reference method under three simulated environmental scenarios. The results are presented and compared to previous field studies. We believe that the laboratory-based method for simulating ambient aerosols presented here could provide in the future a useful alternative to time-consuming and expensive field campaigns, which are often required for instrument certification and calibration.

## 1 Introduction

Atmospheric pollution by airborne particles significantly contributes to climate change and has been linked to respiratory and cardiovascular diseases and lung cancer (Fuzzi et al., 2015; Kim et al., 2015; WHO, 2013). It has been estimated that in Europe alone more than 500,000 deaths per year can be attributed to PM exposure, and that pollution hot spots of PM are responsible for a loss in life expectancy of up to 36 months (Fuzzi et al., 2015). For



EU member states, air quality monitoring - as laid down in the Air Quality Directive 2008/50/EC (European
Parliament, 2008, 2015) – is mandatory and comprises quantification of airborne particulate matter and some of its
constituents. The most important metric to monitor particulate air pollution is the mass concentration, or more
specifically the total mass per unit volume of air of particulate matter which is small enough to pass through a size-
selective inlet with a 50 % efficiency cut-off at 2.5 μm or 10 μm aerodynamic diameter, commonly referred to as
$PM_{2.5}$ and $PM_{10}$ respectively. Ambient limit values for $PM_{2.5}$ and $PM_{10}$ have been established in Europe (European
Parliament, 2008, 2015; FOEN, 2018), the USA (US-EPA, 2016) and other countries worldwide.
Regulatory bodies, air quality networks and atmospheric instrument manufacturers all strive to improve air quality
monitoring, yet there is still a lack of metrological traceability in airborne PM measurements. PM mass
concentration was established as the default metric of PM based on the assumption that mass measurements are
straightforward; they can be performed with a conventional balance. The gravimetric filter-based reference methods
for $PM_{10}$ and $PM_{2.5}$ are set out in the standards EN 12341:2014 (CEN/TC 264/WG-15, 2014) and EN 14907:2005,
however, they fall short in areas such as time resolution and ongoing Quality Assurance and Quality Control to
control the effects of semi-volatile particles and water absorption by particles, for example (CEN/TC 264/WG-15,
2014; Eisner and Wiener, 2002; Hauck et al., 2004; Zhu et al., 2007). The measurement uncertainties for PM mass
concentration in the Directive (European Parliament, 2008, 2015), 25%, are much higher than those for gaseous
pollutants (typically 15%).
Automatic PM monitoring systems were developed in order to avoid these drawbacks and enable time resolutions
below 24 h (Schwab et al., 2006; Weingartner et al., 2011; Zhu et al., 2007); however, demonstrating their
equivalence to the reference manual gravimetric method is time consuming and expensive (Hauck et al., 2004; Zhu
et al., 2007). There are also inconsistencies in the automatic instruments based on different working principles (e.g.
light scattering, beta absorption, oscillating microbalance) and the variations of the aerosols used for comparison.
Ambient PM is not uniform with respect to chemical composition, particle size and shape. In most cases, PM does
not refer to a single pollutant with a distinct chemical signature, but rather to a highly variable mixture of
combustion particles, salts, mineral dust, organic substances and other materials (Hueglin et al., 2005; Putaud et al.,
2010). Therefore, suitable standard calibration aerosols do not currently exist.
To date, automated PM instruments which are used for regulatory purposes (e.g. at national air quality monitoring
stations) are tested for equivalence with the manual gravimetric reference method in monitoring sites using real
ambient air (EC-WG, 2010; Hauck et al., 2004). This requires long and expensive testing campaigns at multiple sites
during different times of the year in an attempt to include all representative meteorological conditions and the
temporal and spatial variations of the ambient air composition. Portable and cost-effective PM monitors, such as the
DustTrak (TSI Inc., USA) and Fidas Frog (Palas, Hermany), which are mostly employed for industrial/occupational
hygiene surveys (Asbach et al., 2018; Davison et al., 2019; Grzyb and Lenart-Boron, 2019), outdoor (Kingham et
al., 2006; Viana et al., 2015; Wallace et al., 2011) and indoor (Chowdhury et al., 2013; Manibusan and Mainelis,
2020; Zhou et al., 2016) air quality investigations, process or emissions monitoring (Al-Attabi et al., 2017; Crilley et
al., 2012; Grall et al., 2018; McNamara et al., 2011) and aerosol research studies, do not necessarily go through
equivalence testing. Instead, they are often calibrated in the laboratory with simple model aerosols, e.g. with dust or





salt particles (Hogrefe et al., 2004; Liu et al., 2017; Papapostolou et al., 2017; Schwab et al., 2004) or dried organic
particles, such as sucrose and adipic acid (Zhang et al., 2018). Such model aerosols, however, are only partially
representative of ambient air since they fail to account for carbonaceous particles and the complex organic matter,
which constitute a considerable mass fraction of airborne particulates (Hueglin et al., 2005; Putaud et al., 2010).
Light-scattering PM monitors are very sensitive to the aerosol size distribution, refractive index (i.e. chemistry) and
humidity, and research findings suggest that a rigorous calibration with "tailored" aerosols, i.e. aerosols
representative of the environment of their intended use, is needed (Jayaratne et al., 2020; McNamara et al., 2011).
The goal of this study was to develop a standardised laboratory-based calibration procedure for automatic
PM-measuring instruments under well-controlled and reproducible experimental conditions. Multi-component
model aerosols were generated in order to reproduce the main properties of real ambient air in terms of particle size
distribution, chemical composition and number/mass concentration, including semi-volatility and hygroscopicity.
The properties of ambient air, of course, may differ dramatically from place to place. Here, the main focus was on
simulating aerosols encountered in Europe (Putaud et al., 2010), which are dominated by organic matter, inorganic
ions (predominantly sulphate and nitrate, and to a lesser extent ammonium), carbonaceous particles (mostly from
fossil fuel combustion rather than biomass burning), mineral dust and water.
Apart from the aerosol generation system, the new setup comprises a flow tube homogeniser and a system for
reference gravimetric measurements. The facility is very versatile: the total PM mass concentration of the model
aerosols can be adjusted in a range from a few $\mu g/m^3$ up to about 500 $\mu g/m^3$, the % fraction of each PM constituent
can be tuned to simulate different urban, suburban or rural aerosols and the aerosol temperature and relative
humidity can be adjusted to simulate winter or summer-like environmental conditions. As a proof of concept, three
different automated PM monitors, the TEOM 1405 (Thermo Scientific, USA), the DustTrak DRX 8533 (TSI Inc.,
USA) and the Fidas Frog (Palas, Germany), were compared with the reference gravimetric method under three
different environmental scenarios. To our knowledge, this is the very first intercomparison involving the Fidas Frog.
Here, we focused on the calibration of the PM monitors' particle quantification, rather than the particle inlet size-
selection; i.e. the TEOM 1405 unit was calibrated without its PM sampling inlet. The Fidas Frog and DustTrak DRX
8533, which are optical instruments, do not possess any size-selective inlet. The facility could be, however, extended
in the future to calibrate PM monitors together with their sampling inlets, if needed. Finally, the facility for
generating ambient-like model aerosols presented in this study is not only relevant for the calibration of PM
monitors but can find applications in the performance evaluation and quality assurance of other aerosol instruments
meant for monitoring ambient, indoor and workplace air as well as in controlled health studies and in vitro
toxicology.
**2 Design and validation of the experimental setup**
The experimental setup consists of three distinct parts: i) the generators of the primary aerosols (dust, salts, soot and
aged soot), ii) a flow tube homogeniser for aerosol mixing, including isokinetic sampling probes and ii) a system for
reference gravimetric measurements. Each part is described in more detail in the following subsections.





### 2.1 Aerosol generation

Four primary aerosols, fresh soot, aged (i.e. organically coated) soot, inorganic salt and mineral dust particles, were generated as depicted in Fig. 1. Fresh soot particles were generated with a miniCAST 6204 burner (Jing Ltd., Switzerland). The operation point was optimised to produce combustion particles with a geometric mean mobility diameter (GMD) of 90 nm and EC/TC (elemental carbon to total carbon) mass fraction of >90 %. The combustion aerosol was split in two portions; one portion was led to the exhaust and the other through a metallic agglomeration tube (1.2 m long, 5 mm internal diameter), where the soot particles grew to about 120 nm. The combustion aerosol was subsequently diluted by a factor of 10 with a VKL10 dilution unit (Palas, Germany). The outlet flow was delivered into an oxidation flow reactor known as Micro Smog Chamber (MSC prototype (Bruns et al., 2015; Corbin et al., 2015b, 2015a; Keller and Burtscher, 2012), developed by A. Keller et al. (Keller and Burtscher, 2012)), where soot was mixed with a controlled amount of α-pinene vapours (≥97 % purity, Sigma Aldrich, Switzerland) under dry conditions (RH<5 %, measured with a digital humidity sensor FHAD 46 series/Almemo D6, Ahlborn, Germany). The aerosol flow through the MSC was set to 1.2 L/min with the use of a miniature radial air blower (model H015X-525A9 with controller, Micronel AG, Switzerland). α-pinene underwent ozonolysis in the MSC, forming secondary organic aerosol (SOA), part of which condensed on the surface of the soot particles, simulating atmospheric ageing procedures (Ess et al., 2020).

The GMD of the soot mobility size distribution was shifted to 160 nm upon coating with SOA and the EC/TC mass fraction dropped to about 20 %. In parallel, fresh soot particles (120 nm mobility diameter) were sampled from the exhaust of the VKL10 dilution unit with the use of a second Micronel blower at flows between 1 and 2 L/min.

Mineral dust particles (ISO 12103-1 A2 fine test dust, Powder Technology Inc., USA) were generated with a rotating brush generator (RBG 1000, Palas, Germany) and were injected horizontally into an empty vessel, which acted as a swirl separator, filtering out the largest size fraction above $PM_{10}$. Alternatively, whenever calibration with respect to the $PM_{2.5}$ faction is desired, a $PM_{2.5}$ impactor can be installed right before injecting the dust particles into the homogeniser.

Inorganic salt particles were generated by nebulising aqueous mixtures of ammonium sulphate and ammonium nitrate at various ratios with the use of a TSI 3076 atomiser (TSI Inc., USA). The particles were passed through a 1.5-m-long, spiral-shaped agglomeration tube to increase the GMD of the (number-based) mobility size distribution to about 100 nm (the mass-based aerodynamic size distribution shows a maximum at ≈200 nm). The aim was to simulate the presence of ammonium, nitrate and sulphate ions in the fine mode of atmospheric particle size distributions (Liu et al., 2000; Wall et al., 1988; Zhuang et al., 1999). Although generation of coarse mode nitrate, formed at coastal areas by the reaction of gas-phase nitric acid with sea-salt or soil dust particles, or coarse mode sulphate was not actively pursued, there is evidence (see Sect. 3) of coarse sulphate formation. Presumably, this is either due to internal mixing of sulphate ions and mineral dust particles in the flow tube homogeniser or to deposition of salt particles in the aerosol pipes and consequent re-entrainment of agglomerates, which are larger than the particles initially produced by the generator.

The primary aerosols were introduced into a flow tube homogeniser (see Sect. 2.2) through separate injection ports. The flow of each primary aerosol entering the homogeniser could be regulated with separate mass flow controllers





(Red-y MFC, Vögtlin, Switzerland) by splitting and directing part of the main primary aerosol flow to the exhaust.
A filter (HEPA capsule, Pall Corporation, USA) was placed upstream of each MFC to remove the particles from the
air flow. All four MFCs were connected to the same aerosol pump (VTE8, Thomas, Germany) as shown in Fig. 1.
The mobility diameter and number concentration of the soot and salt particles were determined with a scanning
mobility particle sizer (SMPS 4.500, Grimm Aerosol Technik GmbH & Co. KG, Germany, L-DMA, Am-241
neutralizer, scan time 695 s). The mass concentration of each primary aerosol was measured with a tapered element
oscillating microbalance (TEOM 1405, Thermo Scientific, USA), operated at a flow rate of 3 L/min and a
temperature of 30 °C. The TEOM data were recorded via a custom-made LabVIEW routine every 6 s without
averaging. The size distribution of the dust particles was measured with a Fidas Frog fine-dust monitor (Palas,
Germany) and a high-resolution optical particle counter LAS-X II (Particle Measuring Systems, USA).

## 2.2 Aerosol homogenisation and sampling

The homogenizer is a 2.3-m-long custom-made stainless steel tube with an inner diameter of 16.4 cm, placed
vertically. The design is based on a previous study, but has been significantly improved and the facility has been
shortened (Horender et al., 2019). The tube is equipped with five identical inlets, placed at the very top as shown in
Fig. 1 and 2(a). Dilution air (filtered, humidity and temperature controlled) is delivered to each one of the inlets at a
flowrate of 24 L/min. The air is conditioned in two steps (Niedermeier et al., 2020) in such a way that the
humidified air is particle free: First, the dew point is adjusted by passing the air through a Nafion humidifier (Series
FC125-240-10MP, PermaPure, USA) filled with water (ultra-analytic grade, Purelab ultra, ELGA, Switzerland) at a
preselected water temperature, adjusted between 3 °C and 30 °C with a cryostat/thermostat (LAUDA Ecoline
Staredition RE 306, Lauda DR. R. Wobser GmbH & Co. KG, Germany). After the Nafion humidifier, the air is fully
saturated with water. Subsequently, the air is guided through a heated hose (Series T-7000, Thermocoax Isopad
GmbH, Germany), where the temperature can be adjusted up to 100 °C. The temperature and RH of the aerosol were
monitored in the homogeniser at the height of the sampling probes with digital sensors (FHAD 46 series/Almemo
D6, Ahlborn, Germany).
The primary aerosols are injected in the middle of the tube through separate ports located 50 cm downstream as
shown in Fig. 2(b). The dilution air sweeps the particles down the tube, where they are further mixed by three
turbulent jets of air. The three air-jet injection tubes (flow rate 20 L/min each) are placed symmetrically around the
homogenizer tube pointing 60° downwards (Fig. 2(b)). The total flow rate of the homogenised aerosol is hence
equal to 180 L/min plus the flows of the four primary aerosols (in total less than 10 L/min). The temperature and
relative humidity of the air-jets are adjusted as described above for the dilution air. Finally, the homogeniser is
surrounded by copper tubes with flowing water in order to maintain the stainless-steel tube at the same temperature
as the aerosol. The temperature of water is adjusted by a flow-type cooler (AS-160 Green Line, Lindr, Czech
Republic) or a thermostat (LAUDA EcoGold E4, Lauda DR. R. Wobser GmbH & Co. KG, Germany). The water
flows in a closed loop, i.e. circulates back to the cryo/thermostat as shown in Fig. 1. Currently, the homogeniser can
only be cooled down to about 10 °C, and this poses limitations to the environmental conditions which can be





simulated in the laboratory; even though the aerosol entering the homogeniser can be preconditioned at a
temperature down to about 5 °C, the aerosol temperature at the outlet of the homogeniser will always be ≥10 °C.
The sampling zone is located 1.25 m downstream of the injection position and accommodates isokinetic sampling
probes (funnels) placed at the bottom end of the homogenizer as illustrated in Fig. 2(c). Isokinetic conditions are
necessary when sampling with instruments operating at different flow rates to ensure representative sampling, e.g.
by minimizing sampling artefacts of larger particles. Several custom-made sampling probes with different cross
sections have been therefore designed to match the flow rate of the various automated PM monitors, which typically
ranges between 0.2 L/min and 20 L/min. It is worth noting that the sampling system is highly adaptable; the lower
end (outlet) of each sampling probe has custom-made threads so that it can be screwed in and out of the bottom
metallic plate of the homogeniser. This ensures that the sampling probes can be readily exchanged before each
experiment depending on the specifications of the PM monitors under test. Finally, the excess aerosol flow exits the
homogeniser through an exhaust outlet connected to a vacuum line as illustrated schematically in Fig. 1.
To characterise the aerosol homogeneity in the flow tube as a function of particle size, sodium chloride (NaCl)
particles with a geometric mean mobility diameter of 50 nm and mineral dust particles with aerodynamic diameter in
the lower μm range (ISO A2 dust) were generated with a nebuliser and a rotating-brush generator, respectively, as
described in Sect. 2.1. Two parallel sampling lines were inserted into the flow tube at the height where the sampling
probes would be normally located; the position of the first sampling line was kept fixed at the centre of the flow tube
(radial position 0) whereas the second one was placed consecutively at a distance $i$ = -70 mm, -50 mm, -30 mm, -10
mm, + 10 mm, + 30 mm, +50 mm and +70 mm with respect to the centre. The outlet of each sampling line was
connected to a calibrated CPC (Models 3775 and 3776, respectively, TSI inc., USA). In total, concentration
measurements at eight different positions along the diameter of the flow tube were performed. The particle number
concentration measured at the centre was used as reference ($C_{ref} = C_0$) and the aerosol homogeneity was calculated
as $C_i/C_{ref}$. The flow rate of each CPC was 0.3 L/min and the inner diameter of the sampling line was 6 mm. This
configuration ensured nearly isokinetic sampling.
The tests were performed with NaCl and mineral dust particles separately. In both cases the aerosol spatial
homogeneity was found to be well within 3 % in number concentration as shown in Fig. 3(a) and (b), respectively,
indicating that the particle mixing characteristics do not depend on particle size in the tested range (i.e. from lower
nm to lower μm range). A final test was performed by mixing NaCl and dust particles to investigate whether the
particle mixing properties are affected when two primary aerosols are introduced into the homogeniser
simultaneously. It was confirmed that the aerosol homogeneity remains well within ±3 % (measurements not
shown), indicating that the simultaneous injection of primary aerosols into the homogeniser through separate ports
(see Fig. 2(b)) does not compromise particle mixing in any way.
By calculating the standard deviation of all 28 measured data points, the spatial inhomogeneity of the aerosol in
terms of number concentration was found to be 1.3 % for coverage factor $k$=1 or 2.6 % for $k$=2. This is used as an
estimate for the uncertainty of the aerosol spatial homogeneity $\eta_{hom}$ (see 4th row of Table 1). This is a crucial
parameter which had not been evaluated so rigorously, if at all, in previous chamber studies (Hogrefe et al., 2004;
Liu et al., 2017; Papapostolou et al., 2017; Schwab et al., 2004; Zhu et al., 2007).



### 2.3 Reference gravimetric method

The reference method used in this study for determining the $PM_{10}$ or $PM_{2.5}$ mass concentrations of particulate matter in the synthetic ambient aerosols is similar to the method described in the standard EN 12341:2014 (CEN/TC 264/WG-15, 2014), i.e. particulate matter was sampled on filters and weighed by means of a balance. The only major deviation from the requirements of the standard is the absence of any size-selective inlets upstream of the automatic PM samplers and the filter holder of the reference gravimetric method.

Briefly, model aerosols were drawn through 47 mm PTFE-coated glass fibre filters (Measurement Technology Laboratories, USA) placed in a metallic filter holder (C806 standard aerosol filter holder, Merck Millipore, Germany). The aerosol flow was controlled with a needle valve and measured with a calibrated mass flow meter (Natec Sensors GmbH, Germany) connected to an aerosol pump (VTE8, Thomas, Germany) in such a way that the volumetric flow corresponded to 2.3 $m^3$/h at ambient conditions. Here, ambient condition refers to the aerosol temperature and pressure in the homogeniser at the height of the sampling probes. In the EN 12341 standard, the requirement that the aerosol flow be set to 2.3 $m^3$/h (=38.33 L/min) at ambient conditions arises from the need to accurately define the size cut-off of the PM inlets, a property that depends on the inlet flow. Since the custom-made facility developed in this study aims at calibrating the PM monitors without their respective PM inlet, this flow requirement is here largely superfluous, apart from effects on sampling from the velocity of air through the filter. Nevertheless, during the experiments the aerosol flow was set to 2.3 $m^3$/h at ambient conditions to facilitate comparison between the conventional field-based and the new laboratory-based procedures. The connecting tube between the isokinetic sampling probe (i.e. central sampling funnel in Fig. 2(c)) and the filter holder was made of inert, electrically conducting rubber material and was kept as short as possible ($\approx$ 5 cm) without bends to minimize deposition losses of particulate matter by kinetic processes as well as losses due to thermal, chemical or electrostatic processes. Finally, the laboratory temperature and pressure were kept constant at (21 ± 1) °C and (950 ± 20) hPa, respectively.

Before sampling, the filters were conditioned and weighed at NPL and shipped in individual plastic containers to METAS. After sampling, the filter samples were placed in Petri dishes, wrapped tightly in plastic cover and stored at 4 °C for about a week. They were then shipped to NPL for conditioning and weighing. NPL use a Measurement Technology Laboratories robotic filter weighing system that comprises an environmental chamber (20 °C ± 1 °C and 47.5 % ± 2.5 % relative humidity), an autohandler system and a Mettler Toledo XP2U balance. The filters are conditioned in the chamber for 48 hours before weighing. The filters are weighed, then the system pauses for 24 hours before reweighing the filters to identify any time-variation in filter mass. Numerous QA/QC checks are made before each set of weighings.

### 2.4 Uncertainty budget for the laboratory-based calibration of PM monitors

The reference mass concentration, $C_{m,ref}$, is given by the equation $C_{m,ref} = \eta_{hom} \frac{m}{V} P_{rel}$, where $\eta_{hom}$ is the aerosol homogeneity in the flow tube, $m$ is the particulate mass collected on the filter and $V$ is the sampled volume. $V$ is given by the aerosol flow through the filter, $Q$, multiplied by the time duration of the measurement $t$. $P_{rel}$ is defined



as the relative particle penetration, $P_{rel} = P_{DUT}/P_{ref}$, where $P_{DUT}$ and $P_{ref}$ is the penetration through the sampling
probe and connecting tube of the device under test (DUT) and the reference method, respectively. The associated
uncertainties are listed in Table 1.
Since sampling is carried out with isokinetic sampling probes and the tubes leading to the filter holder and the DUT
are kept straight and as short as possible, particle losses are minimised. Penetration $P_{rel}$ was set to 1, however, an
uncertainty of 2 % was assigned to account for the higher impaction losses of supermicrometre particles in the
sampling funnel of the reference method due to the higher sampling flow (von der Weiden et al., 2009). These losses
are to some extent counteracted by the lower diffusion losses of submicrometre particles, which decrease with
increasing sampling flow. Here, we followed a rather conservative approach and kept the uncertainty of $P_{rel}$ at 2 %.

## 260 3 Chemical characterisation of model aerosols

Ion chromatography was performed with a Thermo Scientific Dionex™ ICS-1500 Ion Chromatography System for
analysis of Anions and the ICS-2100 model for Cations. The systems consist of a liquid eluent, a high-pressure
pump, an automatic sample injector, a guard and separator column, an electrolytic suppressor, and a conductivity
cell. Before running a sample, the systems were calibrated using a traceable set of calibration standard solutions,
which were prepared in-house. The data produced by the range of calibration standard solutions was used to
calculate calibration coefficients, which were used to quantitate the sample ions.
Thermo-optical analysis of carbonaceous particles was performed with an OC/EC Analyzer (Lab OC-EC Aerosol
Analyzer, Sunset Laboratory Inc., USA), which classified the carbonaceous material as elemental carbon (EC) and
organic carbon (OC). The particles were sampled on quartz fiber filters (Advantec, Tokyo, Japan, QR-100, 47 mm).
For the analysis, the EUSAAR2-protocol (Cavalli et al., 2010) was modified by extending the last temperature step
(850 °C) from 80 s in the original protocol to 120 s in order to ensure complete evolution of carbon (Ess and
Vasilatou, 2019). The charring correction for pyrolyzed OC was performed by transmittance. OC, EC and TC (total
carbon = sum of OC and EC) masses were calculated by the software based on instrument calibration with sucrose
solutions.
The elemental composition of the model aerosols was characterised by combining a cascade impactor for PM
sampling with Total Reflection X-ray Fluorescence Spectroscopy (TXRF, Bruker TStar S4™, Germany) (Osán et
al., 2020). A 13 stage low pressure cascade impactor (Dekati DLPI 10™, Finland) with particle size range from 30
nm to 10 μm was modified to sample at a rate of 10 L/min on smooth and clean commercial-grade acrylic discs with
30 mm diameter, suitable for TXRF. In TXRF, the incident X-ray beam hits the disc's surface at the total reflection
angle. The fluorescence spectrum is detected perpendicular to the surface and is dominated by the contributions
from the deposit, i.e. the sampled particles. This allows for the detection of element masses as low as ≈10 to 100 pg
and thus short sampling periods. The measured element quantities, combined with the sampled air volume, provide
the particle size-selected element mass concentrations in the aerosol. The discs were prepared with a 50 ng Yttrium
standard for TXRF calibration.



As example, the TXRF analysis of model aerosol 1 is shown in Fig. 4. The analysis revealed that the mineral dust
particles contain primarily the elements Si and Al and it was assumed that these are present as oxides $SiO_2$ and
$Al_2O_3$. The mass-based aerodynamic distribution of the $SiO_2$ particles exhibits a maximum in the range 1−2 µm
while the $Al_2O_3$ particles are larger (≈7 µm). Sulphur (i.e. in the form of sulphate ions) appears predominantly in the
submicrometre range (aerodynamic diameter of 30 nm−1 µm) but a second weaker mode is visible at ≈4−7 µm, thus
simulating the aerodynamic size distribution of sulphates in ambient air (Wall et al., 1988; Zhuang et al., 1999)
reasonably well. The coarse mode arises most probably from internal mixing of sulphate ions with mineral dust
particles. Since nitrates and sulphates were generated with the same method, nitrates are expected to exhibit a
similar bimodal size distribution but this could not be experimentally confirmed since nitrogen is difficult to detect
with TXRF spectroscopy. Finally, $K^+$ and $Cl^-$ ions appear in the micrometre range (>2 µm). It is reasonable to expect
that $Na^+$ ions appear also in this size range, however, this could not be investigated by TXRF. By comparing the
results of ion chromatography with those of TXRF spectroscopy, there is no evidence of insoluble potassium.
The results of the chemical analysis of the model aerosols with ion chromatography, EC/OC analysis and TXRF
spectroscopy are summarised in Table 2 and presented graphically in Fig. 5.

## 4 Intercomparision of automated PM monitors with the reference gravimetric method

Three PM monitors, a TEOM 1405 (Thermo Scientific, USA), a DustTrak DRX 8533 (TSI Inc., USA) and a Fidas
Frog (Palas, Germany) were used in this study. The 1405 TEOM takes continuous direct mass measurements of
particulates using a tapered element oscillating microbalance and is considered to be one of the most well-
established automated instruments for monitoring PM mass concentration at air quality monitoring stations. The
DustTrak DRX 8533 and the Fidas Frog aerosol monitors are, unlike TEOM, portable and more cost efficient. These
do not measure particle mass directly but record instead the particle number concentration and size distribution
using optical techniques, from which they calculate the mass concentration using built-in algorithms.
The PM monitors were exposed to three different model aerosols, which were generated in the laboratory with the
facility described in Sect. 2. All three model aerosols were ambient-like mixtures, i.e. they contained inorganic salts,
elemental carbon (soot), secondary organic matter, mineral dust and water. The aerosol composition was analysed
with the methods described in Sect. 3. The chemical composition of the model aerosols and the environmental
conditions during each experiment are listed in Table 2 and depicted schematically in Fig. 5. It can be seen that the
mass fractions of the different chemical constitutents varied in the range ≈30−40 % OM, ≈5−15 % EC, ≈7−15 %
nitrate, ≈5−15 % sulphate, ≈2−3 % ammonium, ≈10−20 % mineral dust and ≈10−20 % other materials.
The $PM_{10}$ mass concentration range (20−40 µg/m³) is typical for urban and suburban regions across Europe. The
chemical composition is representative of European aerosols containing carbonaceous particles from fossil fuel
combustion (rather than biomass burning), secondary organic matter, mineral dust particles and inorganic ions such
as ammonium, sulphate, nitrate and sodium. The temperature and relative humidity of the aerosols were controlled
in the range ≈10−20 °C and 50−70 %, respectively, to simulate different ambient environmental conditions.



The results of the comparison between the automated PM monitors and the reference gravimetric method are shown
in Fig. 6. For the automated PM monitors, which measure continuously and with high time resolution, each data
point corresponds to the arithmetic average over a 30 min measurement period. The reference method delivers only
one data point, i.e. the average $PM_{10}$ mass concentration over the whole measurement period, which is illustrated in
the graph as a straight solid line and summarised in Table 2. It must be noted that the operating temperature of  the
TEOM 1405 monitor was set as low as possible, i.e. to 30 °C, to minimise losses due to (semi)volatile material
(Meyer et al., 2000).  For the DustTrak and Fidas Frog the default factory settings were used.
Figure 6(a) presents the results of the TEOM 1405, Fidas Frog and the reference gravimetric method for model
aerosol 1. The results of the DustTrak 8533 are not reported because of a technical problem (obstruction of the
aerosol inlet) which compromised the measurement accuracy. The TEOM 1405 seems to agree well with the
reference method in the beginning but indicates a decrease of about 15 % in mass concentration at the end of the 4 h
measurement. Particle number concentration measurements of the primary aerosols before and after the experiment
revealed that the number concentration of the fresh soot particles decreased by about 60 % during the measurement
period whereas the number concentration of the dust, salt and aged soot particles remained largely constant. The
reason was a defect in the valve regulating the flow of the fresh soot particles into the homogeniser. The decrease in
the aerosol mass concentration recorded by the TEOM is therefore real and can be attributed predominantly to the
decreasing number and mass concentration of the uncoated soot particles. Since the concentration of the model
aerosol decreased during measurement, the best way to assess the performance of the TEOM 1405 with respect to
the reference method is to calculate the 4-h-average mass concentration. This amounts to 41.6 µg/m$^3$ (see Table 3),
only 3.7 % lower than the reference measurement (43.2 µg/m$^3$).
The fresh soot particles consist mainly of EC and have a geometric mean mobility diameter of about 120 nm, i.e.
below the cut-off limit of the Fidas Frog. Indeed, experiments with miniCAST soot showed that the Fidas Frog and
DustTrak 8533 failed to detect soot particles of this size. This explains why the Fidas Frog reported a constant mass
concentration over the whole measurement period. In Table 3, it can be seen that the Fidas Frog reported an average
$PM_{10}$ mass concentration of 38.8 µg/m$^3$, i.e. -4.4 µg/m$^3$ with respect to the reference method. This deviation agrees
well with the EC mass concentration of 5.0 µg/m$^3$ (Table 2), as determined with EC/OC analysis. Note that the cut-
off curve of optical instruments depends on the refractive index of the particles: the Fidas Frog fails to detect fresh
soot particles below ≈200 nm but detects a consederable mass fraction of the coated soot and salt particles despite
their small size.
The results obtained with model aerosol 2 are displayed in Fig. 6(b). Here, the concentration of the aerosol remained
constant throughout the measurement period. The Fidas Frog and TEOM 1405 monitors underestimate the mass
concentration by 29 % and 14 %, respectively, compared to the reference method while the DustTrak 8533
overestimates the mass concentration by 50 %. The larger deviation between the TEOM 1405 and the reference
method compared to model aerosol 1 results from the winter-like environmental conditions; the temperature of
model aerosol 2 was set to 12 °C, the relative humidity to 70 % and the nitrate content was relatively high (about
15%) as shown in Table 2. Since the aerosol stream sampled by the TEOM 1405 is heated to 30 °C, a fraction of the
(semi)volatile components (e.g. nitrate and secondary organic aerosol) evolves into the gas phase and is therefore



not collected on the filter. These results are in agreement with previous studies reporting that TEOM monitors set at
a lower temperature than the standard configuration (50 °C) still could lose semivolatile materials (Lee et al., 2005),
especially in cooler months (Sofowote et al., 2014; Su et al., 2018).
The large positive deviation of the DustTrak 8533 by a factor of about 1.5 is not surprising. Previous studies have
found that different DustTrak models over-recorded PM values by a factor of 1.2−3 (Chung et al., 2001; Grzyb and
Lenart-Boron, 2019; Heal et al., 2000; Kingham et al., 2006; Liu et al., 2017; McNamara et al., 2011; Wallace et al.,
2011; Yanosky et al., 2002) depending on the aerosol properties. It has been suggested that the "over-estimation is a
simple calibration issue in which differences between the optical properties of the manufacturer's factory calibration
PM (Arizona Road Dust) and the PM under study explained the uniform relative errors recorded" (Kingham et al.,
2006). The results are nevertheless puzzling. Considering that the device fails completely to detect fresh soot and
underestimates the amount of aged soot, we would have rather expected to observe a negative deviation with respect
to the reference method. In any case, the large range of the positive systematic bias (factor of 1.2−3) highlights the
need for source-specific calibration procedures against a reference method.
in the case of Fidas Frog, if the reading of the monitor (21.0 µg/m$^3$, Table 3) is corrected for the undetected mass of
fresh soot (3.8 µg/m$^3$, Table 2), then the Fidas Frog still underestimates the mass concentration by ≈15 % with
respect to the reference method.
The results obtained in the case of model aerosol 3 are illustrated in Fig. 6(c). With an average PM$_{10}$ mass
concentration of 19.2 µg/m$^3$, the TEOM 1405 exhibits an excellent agreement with the reference method (19.3
µg/m$^3$, see Table 2). The DustTrak 8533 overestimates the mass concentration by approx. 33 %, and thus performs
slightly better than in the case of model aerosol 2. Fidas Frog underestimates the mass concentration by about 23 %,
or ≈15 % after correction for the undetected mass of fresh soot, in agreement with the findings of the experiment
with model aerosol 2. As mentioned above, PM monitors based on light scattering, such as the Fidas Frog and the
DustTrak, measure particle number concentration and convert this into mass concentration by using a size-
dependent particle density function. This function is integrated into the software of the instrument. Deviations may
occur if the built-in functions differ substantially from the real density function of the aerosol. More experiments
with ambient-like model aerosols under low and high relative humidity would be needed to define a comprehensive
set of calibration factors for these instruments.

## 5 Conclusions

In this study, we present the first steps towards the generation of ambient-like aerosols in the laboratory. A custom-
made facility for the stable and reproducible generation of such model aerosols was developed, which presents the
following advantages:
• The model aerosols are complex, consisting of elemental carbon (fresh soot), soot coated with SOA (aged
soot), inorganic ions (such as ammonium, sulphate and nitrate) and mineral dust particles
• The aerosol mixture can therefore have a controlled amount of semi-volatile and hygroscopic material



- • The total PM mass concentration of the model aerosols can be adjusted in a range from a few µg/m³ up to
- about 500 µg/m³ and remains stable over several hours
- • The % fraction of each PM constituent can be tuned to simulate different urban, suburban or rural aerosols
- • The size distribution (geometric mean and width of accumulation and coarse mode) can be adjusted by
- tuning the size distribution of the primary aerosols
- • The aerosol temperature and relative humidity can be adjusted to simulate winter or summer-like
- environmental conditions (10−40 °C, 5−90 % RH)
- • A spatial aerosol homogeneity of 2.6 % ($k$=2) in number concentration can be attained in the mixing
- chamber, a parameter not evaluated so rigorously, if at all, in previous chamber studies (Hogrefe et al.,
- 2004; Liu et al., 2017; Papapostolou et al., 2017; Schwab et al., 2004; Zhu et al., 2007)
- • The isokinetic sampling system is highly adaptable and can accommodate instruments with flows up to at
- least 40 L/min
- • The design is much more compact compared to other mixing chambers described in the literature (Hogrefe
- et al., 2004; Horender et al., 2019; Papapostolou et al., 2017; Schwab et al., 2004; Zhu et al., 2007) and can
- therefore easily fit into a typical laboratory.
- As a proof of concept, three different automated PM monitors, the TEOM 1405 (Thermo Scientific, USA), the
- DustTrak DRX 8533 (TSI Inc., USA) and the Fidas Frog (Palas, Germany), were compared with the reference
- gravimetric method under three different environmental scenarios. The TEOM 1405, operated at 30 °C, agreed very
- well with the reference gravimetric method in the case of summertime aerosols (21 °C), but showed a negative
- deviation in $PM_{10}$ mass concentration of ≈15 % when the model aerosol was conditioned at 12 °C due to losses of
- semi-volatile material. The Fidas Frog underestimated the $PM_{10}$ mass concentration by ≈10−30 % whereas the
- DustTrak 8533 overestimated the $PM_{10}$ mass concentration by ≈30−50 % depending on the aerosol chemical
- composition and environmental conditions.
- Currently, one limitation of the facility is that the model aerosols cannot be conditioned to temperatures lower than
- 10 °C but this could be improved by thermally insulating the homogeniser (e.g. with black nitrile foam insulation).
- Moreover, the composition of the model aerosols could be further refined by adding more components, such as
- metallic particles with the use of a spark-discharge generator, bioaerosols e.g. with a Sparging Liquid Aerosol
- Generator (SLAG, CH Technologies, USA) and particles from biomass burning. This last step could pose challenges
- since the mass output is usually not very stable over time and the physicochemical properties of the aerosol depend
- heavily on the combustion material, as well as the stove design.
- To conclude, the facility presented in this study can be used to generate ambient-like model aerosols for quality
- assurance testing, intercomparisons of different instruments and performance evaluation/calibration with respect to
- PM mass concentration. The same facility could also be used for other PM measurements such as number
- concentration and absorption properties (e.g. related to black carbon). The aerosol facility also provides excellent
- opportunities for basic aerosol research and aerosol health-related studies.
- **Data availability**



All data presented in the paper are available for research purposes on request to the authors of the paper.

**Author contribution**
*METAS*: SH and KV designed, validated and operated the experimental facility, coordinated the intercomparison
and prepared the paper with contributions from all other authors; KA designed the isokinetic sampling probes; CCA
assisted during the preparation of the intercomparison and DMK performed EC/OC analysis.
*BAM*: StS performed TXRF analysis
*NPL*: PQ helped design the study, TS weighed the filter samples and KW performed IC analysis
*LNE*: FGL advised on aerosol generation
*DFM*: KD performed high-resolution measurements with a reference optical particle counter
*DTI:* SNS operated the DustTrak DRX during the intercomparison

**Competing interests**
The authors declare that they have no conflict of interest.

**Acknowledgments**
S. Horender, K. Auderset and K. Vasilatou would like to thank their colleagues at the mechanical and electronic
workshop (METAS) for valuable technical assistance throughout this study.
This work has received funding from the 16ENV07 Aeromet project of the European Union through the European
Metrology Programme for Innovation and Research (EMPIR). EMPIR is jointly funded by the EMPIR participating
countries within EURAMET and the European Union. METAS was supported by the Swiss State Secretariat for
Education, Research and Innovation (SERI) under contract number 17.00112. The opinions expressed and
arguments employed herein do not necessarily reflect the official views of the Swiss Government.

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



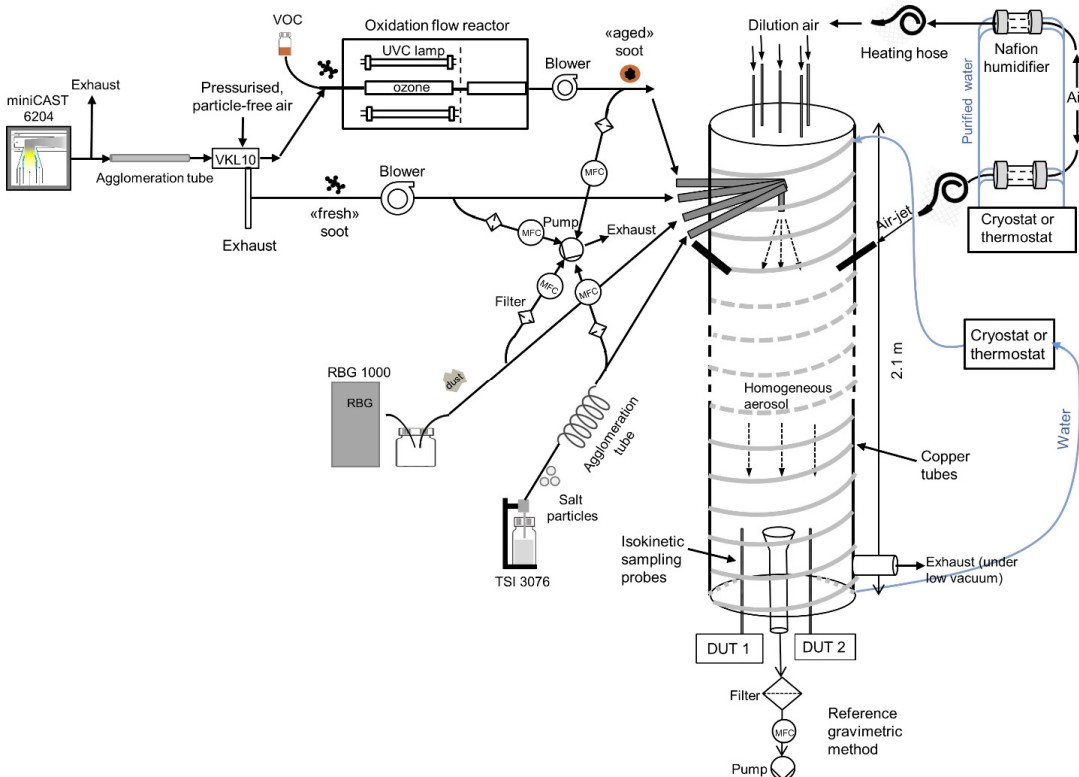


**Figure 1: Schematic illustration of the experimental setup. DUT stands for device under test.**





Inlets for
dilution air

(a)

Inlets for
primary
aerosols

(b)

Inlet for
turbulent
air-jet

(c)

Filter holder
for gravimetric
method

Connecting tube
for PM instrument


**Figure 2: a) Computer-aided design (CAD, Inventor Professional 2019, Autodesk, USA) of the homogeniser. Panels (b)**
**and (c) show enlarged views of the primary aerosol inlets and isokinetic sampling probes, respectively.**





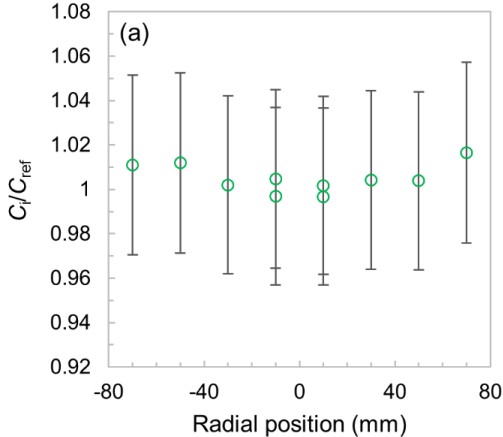
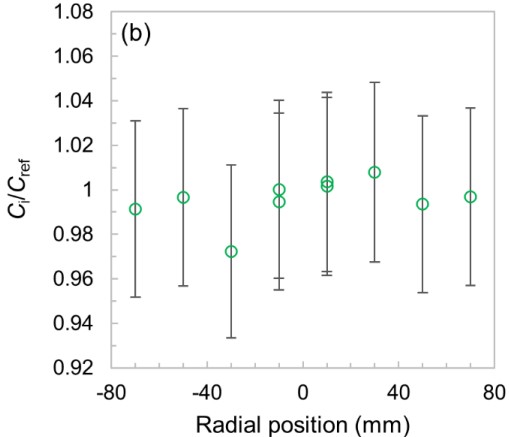

**Figure 3: Aerosol spatial homogeneity, $\eta_{hom} = C_i/C_{ref}$ , at various radial positions along the diameter of the flow tube with a) NaCl (sodium chloride) and b) mineral dust particles as test aerosols. The measurements at positions $i = $ -10 mm and + 10 mm were performed twice to assess measurement reproducibility. The error bars designate expanded uncertainties (95 % confidence level). These are type B uncertainties from the combined measurement uncertainties of the two CPCs and have no influence on the determination of homogeneity since they would shift all data points up or downwards by the same amount.**





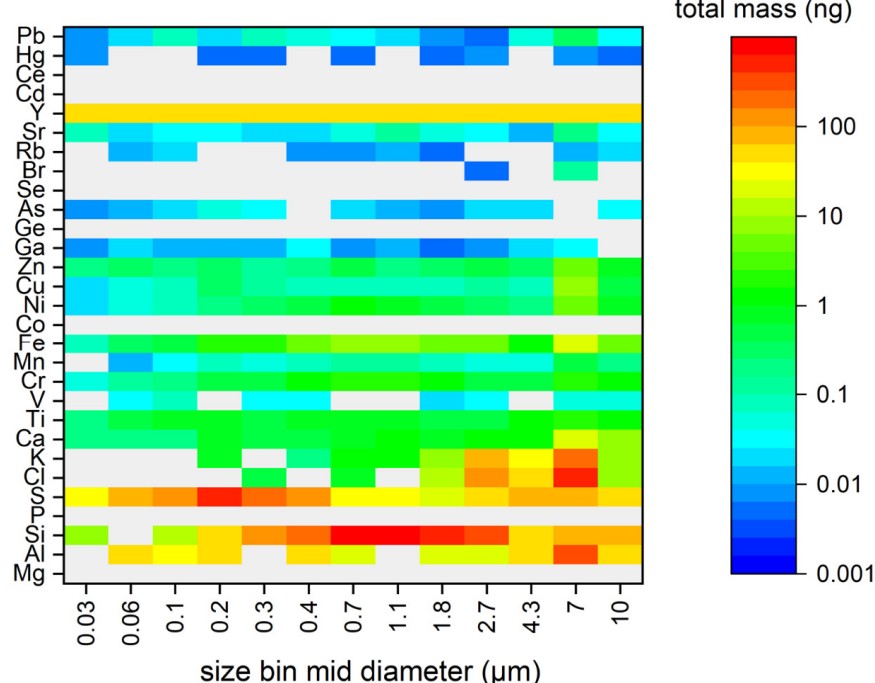

**Figure 4: TXRF analysis of model aerosol 1 (see text and Table 2 for a discussion on all three model aerosols).**

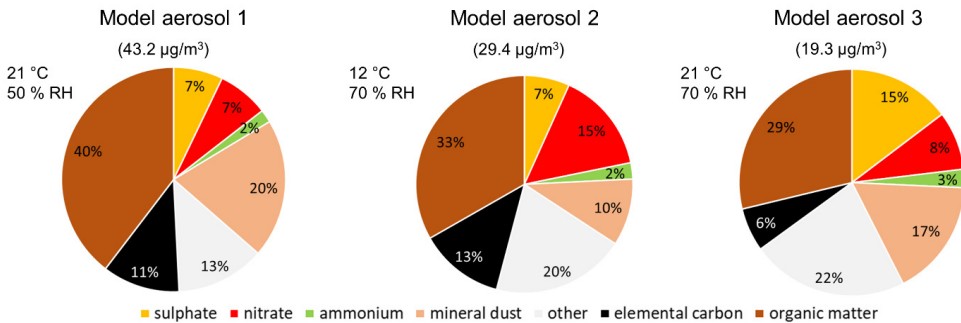

**Figure 5: PM composition (%) of the three model aerosols and environmental conditions during each experiment.**



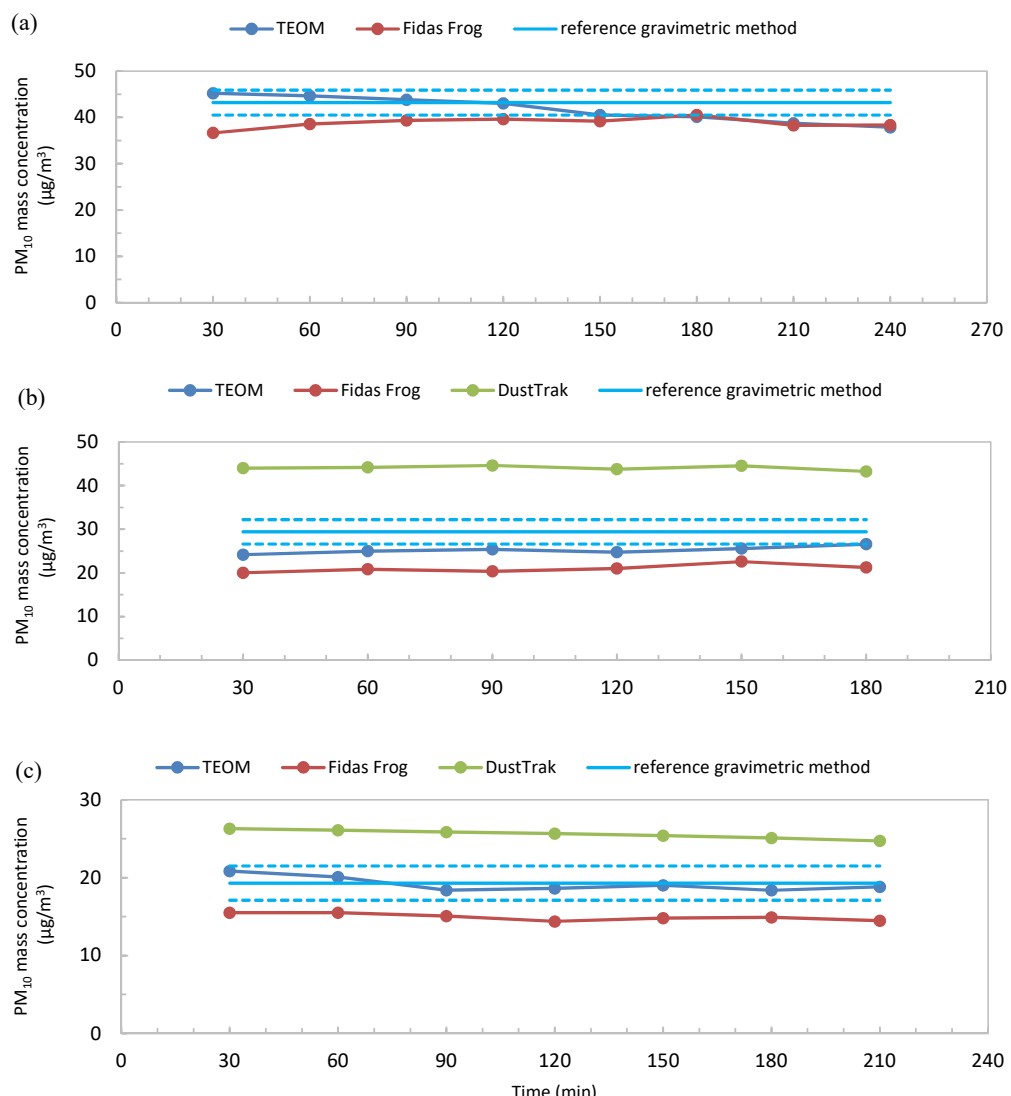

**Figure 6: PM$_{10}$ mass concentrations reported by the TEOM 1405, DustTrak DRX 8533 and Fidas Frog monitors compared to the results of the reference gravimetric method in the case of a) model aerosol 1, b) model aerosol 2 and c) model aerosol 3. In Fig. 6(a), the results of the DustTrak 8533 are not plotted because of technical issues during measurement (see text for more details). The dashed lines designate the expanded uncertainties (95% confidence level) of the reference PM$_{10}$ value.**



**Table 1: Example of the uncertainty budget for a PM$_{10}$ mass concentration of 40 µg/m³ and a sampling time of 240 min.**

| Quantity | Value (example) | Standard uncertainty ($k$=1) | Relative uncertainty (95 % confidence level) |
|---|---|---|---|
| $t$ | 240 min | negligible | negligible |
| $P_{\mathrm{rel}}$ | 1.00 | 0.01 | 2 % |
| $\eta_{\mathrm{hom}}$ | 1.000 | 0.013 | 2.6 % |
| $Q$ | 38.333 L/min | 0.058 L/min | 0.30 % [1] |
| $m$ | 368.0[2] µg | 8.4 µg | 4.6 % |
| $C_{m,\mathrm{ref}}$ | 40.00 µg/m³ | 1.13 µg/m³ | 5.7 % |

[1] The mass flow meter (Natec Sensors GmbH, Germany) was calibrated at METAS in a traceable manner. The expanded relative uncertainty on the calibration certificate amounts to 0.15 %. Here, a conservative estimation of 0.30 % was made to

account for possible drifts since the time of calibration.

[2] Assuming no loss of particulate mass during filter conditioning.





**Table 2: Chemical composition of the three model aerosols, mass concentration ($\mu g/m^3$) of each chemical constituent and**
40 **environmental conditions during each experiment.**

| Model aerosol | Sulphate ($\mu g/m^3$) | Nitrate ($\mu g/m^3$) | Ammonium ($\mu g/m^3$) | Mineral dust ($\mu g/m^3$) | EC [1] ($\mu g/m^3$) | OC [1] ($\mu g/m^3$) | OM [2] ($\mu g/m^3$) | Other [3] ($\mu g/m^3$) | T (°C) | % RH |
|---|---|---|---|---|---|---|---|---|---|---|
| 1 | 3.06 ± 0.13 | 3.17 ± 0.11 | 0.80 ± 0.12 | 8.6 ± 2.6 | 4.8 ± 0.6 | 10.0 ± 0.8 | 17.0 ± 3.4 | 5.5 ± 0.2 | 21 ± 1 | 50 ± 2 |
| 2 | 2.03 ± 0.09 | 4.53 ± 0.16 | 0.73 ± 0.20 | 3.0 ± 0.9 | 3.8 ± 0.5 | 6.0 ± 0.5 | 10.2 ± 2.0 | 6.0 ± 0.2 | 12 ± 1 | 70 ± 3 |
| 3 | 3.07 ± 0.12 | 1.75 ± 0.11 | 0.55 ± 0.10 | 3.5 ± 1.1 | 1.3 ± 0.2 | 3.6 ± 0.3 | 6.1 ± 1.2 | 4.7 ± 0.2 | 21 ± 1 | 70 ± 3 |

[1] The reported uncertainties do not include uncertainties in the determination of the split point.

[2] In past studies with atmospheric aerosols, factors between 1.1 and 2.1 have been proposed to convert OC to OM mass (El-
45 Zanan et al., 2005). The Micro Smog Chamber is known to yield moderately to strongly oxidised secondary organic matter
(Bruns et al., 2015), thus a factor of 1.7 ± 0.3 was assumed.

[3] Mostly $Na^+$ and to a lesser extent $K^+$ and $Cl^-$ from contamination of the aerosol generation system and, possibly, impurities
in the mineral dust mixture.



**Table 3: Average PM$_{10}$ mass concentration (µg/m$^3$) reported by the TEOM 1405, Fidas Frog and DustTrak 8533 automated PM**
**monitors and the referene gravimetric method.**

| | Average PM$_{10}$ mass concentration (µg/m$^3$) | | | |
|---|---|---|---|---|
| Model aerosol | TEOM 1405 | Fidas Frog | DustTrak 8533 | Reference gravimetric method |
| 1 | 41.6 | 38.8 | -[1] | 43.2 ± 2.7 |
| 2 | 25.3 | 21.0 | 44.0 | 29.4 ± 2.8 |
| 3 | 19.2 | 15.0 | 25.6 | 19.3 ± 2.2 |

[1]The result was discarded because of a technical issue during measurement.