# Peer review of "Facility for generation of ambient-like model aerosols in the laboratory: application in the intercomparison of automated 3 PM monitors with the reference gravimetric method"

_Atmospheric Measurement Techniques, 2020_

## Referee Comment (RC1) · Anonymous Referee #1 · 19 Nov 2020

Designing an aerosol generator to simulate ambient-like aerosols is a complex task especially because most ambient chemical reactions are hard to characterize. However, they are extremely important for instrument detection abilities. The authors have thoroughly explained the setup, its advantages and its limitations. There are a few improvements that can be made to the outputs of this setup, but this work is important to get out which can then be further developed. Calibrating instruments based on the conditions in which they are used and with relevant aerosol characteristics is very important, especially as we start to understand the importance of particle coatings, aggregation, etc. on human health and climate interactions. This type of device will also be important as wearable technology and supersites become more common. An over or under estimation of the particle concentration could prevent health alerts from going out if underestimated or could cause unnecessary concern if there is a consistent over estimation of PM. This is a useful proof of concept paper and provides a useful path forward in developing this device, and its uses, further.

Specific comments: Are there studies that look at each of the instrument's reaction to the ambient chemical on their own and not aggregated into a standardized aerosol? Line 367 alludes that this needs to be done and is an evidence gap in the literature. Is this true?

This device seems like a nice way to incorporate more ambient aerosols into calibrating instruments, but there seems to be relatively large unknowns such as needing to understand each component's influence on the instrument separately and how particle density influences instrumentation response. Will this presumably expensive aerosol generator have improvements to the aerosol modelling field even with these unknowns? Can this be corrected for during post-processing?

What is the lowest sample flow that this device can adhere to?

Would this type of system require knowledge about the ambient chemical composition in order to tailor the device to a calibration setting relevant to the geographical area? If so, to what degree is required for accurate results?

Can the designed device be taken out into the field for calibration outside of the lab? Or does it require a built-in power source, lots of set up or a highly stable environment (such as a study platform, etc.). In figure 6b, there are two different injection tubing types (L-shape and a bent elbow shape). What was the reason for the difference in this design? Does the L-shape influence the incoming primary aerosols differently than the turbulent air jet?
Most of these parts are custom made and look very expensive. I assume that this device is going to be patented and sold, but in the case that it is not, how will other researchers be able to recreate these results? Will it be obtainable for using in a low-cost setting?

Technical Corrections: Line 37 – "most important metric to monitor", should be changed to "regulated pollutant" or something similar, as health studies are not conclusive about what particle characteristic is the most damaging to human health (PM, particle number, surface area, surface chemical composition, etc.). PM2.5 and PM10 is significantly easier to measure than surface area or PN for fine particles. Line 47 – Could the authors give a quick description of what the references found in regards to the limitations of measuring PM due to the volatile component of particles?

Throughout – "PM mass" is redundant, as PM stands for particulate mass.

Line 49 – Is this last sentence saying that PM has a 25% uncertainty with the measurement techniques? This sentence could be structured differently to avoid confusion to the reader. Line 60 – would the standard to which instruments need to be standardized be different depending on where/when/how PM is being measured? What about the influence of elevation in this standardization process? Line 87 – Add "detailed below" after "aerosol generator system" so it reads: Apart from the aerosol generator system (detailed below), ... Line 87- Refers to a "new setup". Is there an old setup that precedes this setup that should be referenced? Line 105 – roman numeral should be iii) at the end of the line. Line 111 – was there a specific reason for choosing GMD of 90 nm? Line 113 – Was the aggregated particle size actually measured or is this an assumption based off of particle growth theories? Line 119 – Is this mixing protocol to adhere gaseous particles to the soot particles? A brief sentence about this would be helpful.

Figure 1 is very nice. It would be helpful to emphasize the intake air. How it is currently makes it look like a closed loop system rather than drawing air into the system. Is the
recycled water from the aerosol mixture chamber just used for cooling, is it cleaned for reuse, or is that unnecessary?

Line 272 – what is the purpose of the charring correction? Line 285 – Missing "an" before "example" Line 306 – Here would be a good place to include the uncertainty of these PM calculation technique (+/- 20%?). Line 332 – Is there any possibility the fresh soot was aggregating which caused a decrease in PN? Or was this variable accounted for in the design? Line 369-371 – is this sentence out of place? It seems relevant to include it with the previous paragraph.

---

## Author Comment (AC1) · 25 Nov 2020

We would like to thank Reviewer 1 for the positive comments and feedback that have helped to improve the manuscript. Below please find the response to each comment as well as the sections of the manuscript that were adapted.

Reviewer 1: Are there studies that look at each of the instrument's reaction to the ambient chemical on their own and not aggregated into a standardized aerosol? Line 367 alludes that this needs to be done and is an evidence gap in the literature. Is this

true?

Answer: It is true that more experiments are needed to characterise the instruments' response to single-component aerosols, especially in the mobility size range 150 nm 300 nm where the lower cut-off of the efficieny is expected to be in the case of the light-scattering instruments.

Reviewer 1: This device seems like a nice way to incorporate more ambient aerosols into calibrating instruments, but there seems to be relatively large unknowns such as needing to understand each component's influence on the instrument separately and how particle density influences instrumentation response. Will this presumably expensive aerosol generator have improvements to the aerosol modelling field even with these unknowns? Can this be corrected for during post-processing?

Answer: This manuscript is meant as a proof of concept that synthetic ambient-like aerosols can be produced in the laboratory and used for instrument calibration. We agree with the Reviewer that a systematic characterisation of the instruments' response with respect to each aerosol chemical component/density would be valuable and we intend to pursue such a study in the near future.

Reviewer 1: What is the lowest sample flow that this device can adhere to?

Answer: The aerosol flow through the homogeniser is about 180 L/min (120 L/min dilution air + 60 L/min from the three turbulent air jets + 2-5 L/min from the different aerosol generators). The devices under test, e.g. PM monitors, to be connected at the bottom part of the homogeniser can have a sample flow ranging from a few hundred mL/min up to about 40 L/min.

Reviewer 1: Would this type of system require knowledge about the ambient chemical composition in order to tailor the device to a calibration setting relevant to the geographical area? If so, to what degree is required for accurate results?

Answer: Yes, the calibration aerosol needs to simulate the chemical properties of the
specific ambient aerosol(s) in the location where the PM monitor will be installed. In other words, it is important to know the average % mass fraction of elemental carbon, organic mass, dust and inorganic ions of the ambient aerosol (with an uncertainty of about 30% or, if possible, better), so that the composition of the calibration aerosol can be tuned accordingly. The design of the experimental facility will remain the same, but the set-points of the different aerosol generators will need to be adjusted.

Reviewer 1: Can the designed device be taken out into the field for calibration outside of the lab? Or does it require a built-in power source, lots of set up or a highly stable environment (such as a study platform, etc.). In figure 6b, there are two different injection tubing types (L-shape and a bent elbow shape). What was the reason for the difference in this design? Does the L-shape influence the incoming primary aerosols differently than the turbulent air jet?

Answer: The current facility is not transportable, but we are currently working on the miniaturisation of the flow tube homogeniser in order to reduce the overall dimensions of the set up. The idea behind the design of this facility was to perform laboratory-based calibrations in order to avoid the laborious field campaigns. In Figure 2b), the L-shaped inlets are meant for the primary aerosols which are delivered centrally and downwards in the homogeniser. The inlets with a 60 degree angle are specially designed for the turbulent air jets. The 60 degree angle helps to create vortices which are necessary for an efficient mixing of the different aerosol components.

Reviewer 1: Most of these parts are custom made and look very expensive. I assume that this device is going to be patented and sold, but in the case that it is not, how will other researchers be able to recreate these results? Will it be obtainable for using in a lowcost setting?

Answer: Indeed, some parts of the facility (e.g. homogeniser and isokinetic sampling probes) are custom-made. The material costs less than 2'000 € in total, but the construction must be carried out by a trained technician. The facility will not be patented,

which explains why we actually present the CAD-design of the homogeniser in the manuscript. Our aim is to share our know-how with the community, receive feedback and improve the design of the facility even further. Other researchers are free to reproduce the facility as long as they cite the current manuscript.

Reviewer 1: Line 37 – "most important metric to monitor", should be changed to "regulated pollutant" or something similar, as health studies are not conclusive about what particle characteristic is the most damaging to human health (PM, particle number, surface area, surface chemical composition, etc.). PM2.5 and PM10 is significantly easier to measure than surface area or PN for fine particles.

Answer: The reviewer is absolutely right. The sentence now reads: "most important regulated metric to monitor...".

Reviewer 1: Line 47 – Could the authors give a quick description of what the references found in regards to the limitations of measuring PM due to the volatile component of particles?

Answer: Concerning the TEOM, the larger the percentage of volatile material the higher the risk of material loss on the filter of the TEOM. The losses depend of course on the ambient temperature and the temperature of the TEOM's measurement cell, with the losses being higher when TTEOM» Tambient (see, for instance, Mayer et all 2000, Lee et al. 2005, Sofowote et al. 2014 and Su et al. 2018 cited in the manuscipt). Concerning the PM monitors based on light scattering, we believe that it's mostly the hygroscopic growth of particles that could create measurement artefacts, that's why PM monitors installed at air quality monitoring stations are typically equipped with a drier or a heated inlet.

Reviewer 1: Throughout – "PM mass" is redundant, as PM stands for particulate mass.

Answer: PM can either stand for Particulate Matter (see, for instance, definition of PM2.5 and PM10) or Particulate Mass. In the manuscript (abstract), we have defined

PM as Particulate Mass. If the Reviewer agrees, we would like to keep the term PM mass.

Reviewer 1: Line 49 – Is this last sentence saying that PM has a 25% uncertainty with the measurement techniques? This sentence could be structured differently to avoid confusion to the reader.

Answer: The Reviewer is right. We have restructured the sentence: "The measurement uncertainties for PM mass concentration in the Directive (European Parliament, 2008, 2015) are 25%, and thus much higher than those for gaseous pollutants (typically 15%)".

Reviewer 1: Line 60 – would the standard to which instruments need to be standardized be different depending on where/when/how PM is being measured? What about the influence of elevation in this standardization process?

Answer: The calibration aerosol needs to simulate the properties of the real ambient aerosol(s) at the location in which the PM monitor will be installed. This means that the calibration aerosol must be realistic in terms of both chemical composition and size distribution. Concerning th question about "elevation", we believe that most PM monitors have a temperature and pressure sensor to calculate the sampled volume.

Reviewer 1: Line 87 – Add "detailed below" after "aerosol generator system" so it reads: Apart from the aerosol generator system (detailed below), . . .

Answer: We have modified the text according to the Reviewer's suggestion. The sentence now reads: "Apart from the aerosol generation system (detailed below), ...".

Reviewer 1: Line 87- Refers to a "new setup". Is there an old setup that precedes this setup that should be referenced?

Answer: Here, we just wanted to highlight that the setup is novel (and does not rely on a previous facility).

Reviewer 1: Line 105 – roman numeral should be iii) at the end of the line.

Answer: We thank the reviewer for spottting this mistake. We have corrected the text accordingly.

Line 111 – was there a specific reason for choosing GMD of 90 nm?

Answer: The GMD was chosen to be 90 nm to simulate soot particles emitted by diesel vehicles.

Reviewer 1: Line 113 – Was the aggregated particle size actually measured or is this an assumption based off of particle growth theories?

Answer: The particle size (i.e. mobility diameter of 120 nm) was measured with a Scanning Mobility Particle Sizer (SMPS). We have added a clarification in the text.

Reviewer 1: Line 119 – Is this mixing protocol to adhere gaseous particles to the soot particles? A brief sentence about this would be helpful.

Answer: The aerosol flow through the MSC was set to 1.2 L/min to allow enough time for the ozonolysis of the $\alpha$-pinene and subsequent condensation of the SOA onto the soot surface. Higher aerosol flows through the MSC would lead to a too short residence time in the reactor and should be avoided. We have added an explanation in the text.

Reviewer 1: Figure 1 is very nice. It would be helpful to emphasize the intake air. How it is currently makes it look like a closed loop system rather than drawing air into the system. Is the recycled water from the aerosol mixture chamber just used for cooling, is it cleaned for reuse, or is that unnecessary?

Answer: We thank the Reviewer for the positive feedback. The aerosol does not flow in a closed loop. The dilution air is introduced at the top of the homogeniser and the primary aerosols are injected a few cm downwards. The aerosol is homogenised, part of it is sampled by the PM monitors and the reference gravimetric method, and the rest exits the homogeniser through the exhaust outlet (see explanation in Lines 191-

192). In Figure 1, the exhaust outlet is shown close to the bottom right edge of the homogeniser. The water used for cooling (or heating) of the homogeniser does not need to be cleaned but needs to be replenished every few weeks depending on the hours of operation.

Reviewer 1: Line 272 – what is the purpose of the charring correction?

Answer: When organic carbon undergoes charring, it could mistakenly be classified as elemental carbon. A charring correction is performed to correct for this artefact and improve the accuracy of the EC/OC split in thermal-optical methods.

Reviewer 1: Line 285 – Missing "an" before "example"

Answer: We have corrected the text. The sentence now reads: "As an example, ...".

Reviewer 1: Line 306 – Here would be a good place to include the uncertainty of these PM calculation technique (+/- 20%?).

Answer: The measurement uncertainty depends on the composition of the aerosol, the environmental conditions and the calibration technique used by the manufacturer. The uncertainties can be much higher than 20% (deviations up to a factor of 3 from the reference gravimetric method have been reported in the literature).

Reviewer 1: Line 332 – Is there any possibility the fresh soot was aggregating which caused a decrease in PN? Or was this variable accounted for in the design?

Answer: The size of the soot particles was monitored with an SMPS and was shown to remain stable. We believe that the decrease in the soot particle number concentration was due to the decreasing flow of the soot aerosol injected into the homogeniser.

Reviewer 1: Line 369-371 – is this sentence out of place? It seems relevant to include it with the previous paragraph.

Answer: Since "the large range of the positive systematic bias (factor of $1.2-3$)..." refers specifically to the DustTrak and not the TEOM, we cannot shift the sentence to

the previous paragraph. If the Reviewer agrees, we would prefer to leave the sentence at the current position.

---

## Referee Comment (RC2) · Anonymous Referee #2 · 4 Dec 2020

Horender et al. describe a facility for the generation of model aerosols in the laboratory from a range of primary and secondary constituents. They describe and characterise aerosol generation, particle conditioning and homogenisation, and sampling probe design. The system is impressively engineered and carefully characterised, and an example application which compares aerosol mass measurements from three devices with a reference gravimetric technique shows proof-of-concept. Improved characterisation of aerosol mass sensors and other particle measurements is an important topic and I enjoyed reading the manuscript. I recommend publication after the following points are

addressed.

General comments: I have one major comment: The strategy of generating ambient-like particle mixtures in the lab is more convenient and reproducible than a field campaign, but seems to have the same main weakness for validation and calibration – it is difficult to understand which aerosol component(s) and properties (size, shape, composition...) are responsible for discrepancies between instruments, and hence how to make improvements. For example, here the authors are puzzled why the Dust-Trak overestimates mass.

Aside from the SOA-coated soot, the aerosol population in the flow tube is an external mixture. For the current application to particle mass sensors I think it would be easier and more conclusive to test with each particle type separately? But given the emphasis of the current paper, what role do the authors see for using ambient-like aerosol mixtures beyond proof of concept?

Specific comments:

Title/abstract: Would the authors like to give a name/acronym to the facility? This may sound like a strange suggestion but could help with adoption and referencing in future!

Line 18 and line 115: please clarify photo-oxidation vs ozonolysis. What is the light wavelength in the Micro Smog Chamber? Which aging processes (ozonolysis, direct photolysis, OH oxidation etc...) occur under these conditions?

Line 25: In reference to Review 1, I support the authors' definition of "PM" as particulate matter and subsequent use of the term "PM mass".

Line 117: How is alpha-pinene delivery controlled? Is the concentration monitored or can an estimate be provided?

Line 155: The tube is marked as 2.1m in Fig 1. Please clarify.

Line 212: Please define "coverage factor".

Line 313: 10-20% "other material" is attributed to contamination or mineral coatings in Fig 5. This seems concerningly high given the controlled aerosol production. What purities of ammonium sulfate/nitrate and water were used in the atomiser? Are the collection substrates or other aspects of aerosol chemical analysis potential contamination sources? If the mineral dust is analysed separately, how much "other material" is identified?

Line 326-382: I suggest this discussion and Fig 6 be re-ordered to start with the currently labelled aerosol 2 and 3. The discussion around aerosol 1 is confused by technical problems with the DustTrak and soot source. In general the composition of the three model aerosols seems similar and it's unclear if we are meant to focus on any differences? The main differences seem to be RH and temperature.

Line 355: The role of hygroscopicity and aerosol liquid water in influencing the performance of particle mass sensors (e.g. Crilley et al., 2018; Di Antonio et al, 2018) should be mentioned. The evaporation of water in the TEOM due to heating may be at least as important as nitrate/organic matter. Liquid water is a potential source of difference for all the online methods (70% RH) when comparing to the gravimetric reference (47.5% RH) especially if hygroscopic components effloresce on the filter. In future, the water content of the aerosol could be estimated from composition, RH and temperature using a thermodynamic model such as E-AIM (http://www.aim.env.uea.ac.uk/aim/aim.php).

Line 369: Capitalise "in".

Line 390: The data in the paper focuses on 20-40 ug/m3. It's hard to conclude from this that the stable operating range is "few to 500 ug/m3". Please include some additional evidence or modify this statement.

Figure 6: Please use different marker shapes as well as colours for the different instruments.

References: Crilley et al., AMT, 11, 709–720, 2018. Di Antonio et al., Sensors, 18(9),

2790, 2018.

---

## Author Comment (AC2) · 16 Dec 2020

We would like to thank Reviewer 2 for the valuable feedabck, discussion and comments that have helped us to improve the quality of the manuscript. Below please find the point-by-point response to the Reviewer's comments as well as the sections of the manuscript that were modified.

Reviewer 2: General comments: I have one major comment: The strategy of generating ambient-like particle mixtures in the lab is more convenient and reproducible than

a field campaign, but seems to have the same main weakness for validation and calibration – it is difficult to understand which aerosol component(s) and properties (size, shape, composition. . .) are responsible for discrepancies between instruments, and hence how to make improvements. For example, here the authors are puzzled why the DustTrak overestimates mass. Aside from the SOA-coated soot, the aerosol population in the flow tube is an external mixture. For the current application to particle mass sensors I think it would be easier and more conclusive to test with each particle type separately? But given the emphasis of the current paper, what role do the authors see for using ambient-like aerosol mixtures beyond proof of concept?

Answer: We believe that we should distinguish between i) instrument characterisation and ii) instrument calibration.

i) We agree with the Reviewer that experiments with single-component aerosols would be valuable for characterising certain aspects of the measuring instruments, such as the 50% cut-off at small particle sizes (150 nm - 300 nm) in the case of light-scaterring detectors, which depends on material properties. Such experiments can easily be performed with the facility presented here by simply switching on the aerosol generator of interest (and keeping all other generators switched off).

ii) For calibrating PM monitors with respect to PM mass concentration, however, the use of single-component aerosols (e.g. dry mineral dust) would be unrealistic. It is known that the TEOM suffers from artefacts related to losses of (semi)volatile material. The light-scattering PM monitors suffer from artefacts due to hygroscopic growth of the particles and from biases related to the built-in algorithms which convert the measured particle size distributions into mass concentrations by assuming a size-dependent particle density function. Therefore, for the PM mass calibration to be meaningful, the calibration aerosol needs to comprise the right fraction of hygroscopic and volatile compounds, exhibit a realistic particle size distribution (accumulation and coarse mode) and a realistic size-dependent density. This can only be achieved by generating complex ambient-like aerosols in the laboratory. In the last three months, i.e. since the

manuscript was posted online, we have already been able to go beyond "proof-of-concept". We have used complex aerosol mixtures to characterise a newly developed Black Carbon monitor with aerosols of different single scattering albedo, and calibrate a commercial PM low-cost sensor as well as two light-scattering PM monitors against the reference gravimetric method.

Reviewer 2: Title/abstract: Would the authors like to give a name/acronym to the facility? This may sound like a strange suggestion but could help with adoption and referencing in future!

Answer: We agree with the Reviewer's suggestion and we have changed the title to "Facility for production of ambient-like aerosols (PALMA) in the laboratory: application in the intercomparison of automated PM monitors with the reference gravimetric method". We have made a similar change in the "Abstract" and "Conclusions" of the manuscript.

Reviewer 2: Line 18 and line 115: please clarify photo-oxidation vs ozonolysis. What is the light wavelength in the Micro Smog Chamber? Which aging processes (ozonolysis, direct photolysis, OH oxidation etc. . .) occur under these conditions?

Answer: The UVC lamps in the MSC emit at $\lambda$=254 nm and 185 nm. Since the relative humidity in the photo-oxidation reactor remains <5 % (see Section 2.1), the main ageing process is the ozonolysis of $\alpha$-pinene. The generation of OH radicals is negligible at such low RH. In the abstract, we have made the following clarification: "Model aerosols containing elemental carbon, secondary organic matter from the ozonolysis of $\alpha$-pinene...".

Reviewer 2: Line 117: How is alpha-pinene delivery controlled? Is the concentration monitored or can an estimate be provided?

Answer: We have added the following clarification in the text: "The concentration of $\alpha$-pinene at the inlet of the MSC was determined with a photoionization detector (PID

PhoCheck TIGER, Ion Science Ltd, UK) after filtering out the particles. The concentration could be varied by adjusting the flow of air through the $\alpha$-pinene container (gas bubbler) and typically ranged between 60-70 ppm".

Reviewer 2: Line 155: The tube is marked as 2.1 m in Fig 1. Please clarify.

Answer: The length of the tube is indeed 2.1 m and therefore Fig. 1 is correct. We have corrected, however, the first sentence of Section 2.2 as follows: "The homogenizer is a 2. 1-m-long custom-made stainless steel tube with an inner diameter of 16.4 cm". Thank you to the Reviewer for pointing out this inconsistency between Figure 1 and the text in Section 2.2.

Reviewer 2: Line 212: Please define "coverage factor".

Answer: Thank you to the Reviewer for pointing this out to us. The sentence now reads: "By calculating the standard deviation of all 28 measured data points, the spatial inhomogeneity of the aerosol in terms of number concentration was found to be 1.3 % for coverage factor k=1 (i.e. 68 % confidence level) or 2.6 % for k=2 (i.e. 95 % confidence level)."

Reviewer 2: Line 313: 10-20% "other material" is attributed to contamination or mineral coatings in Fig 5. This seems concerningly high given the controlled aerosol production. What purities of ammonium sulfate/nitrate and water were used in the atomiser? Are the collection substrates or other aspects of aerosol chemical analysis potential contamination sources? If the mineral dust is analysed separately, how much "other material" is identified?

Answer: We believe that the major source of impurities is contamination of the flow tube homogeniser with NaCl from previous experiments (aerosol spatial homogeneity studies, Figure 3). Although we made sure to flush the tube with clean air for several hours before each experiment, it seems there were still NaCl particles deposited in the aerosol inlet. By cleaning meticulously the aerosol inlet with wet wipes, it is possible to keep the mass fraction of "other material" well below 10%. We have added a clarification in Table 2.

Reviewer 2: Line 326-382: I suggest this discussion and Fig 6 be re-ordered to start with the currently labelled aerosol 2 and 3. The discussion around aerosol 1 is confused by technical problems with the DustTrak and soot source.

Answer: The Reviewer is right that the experiments related to Aerosol 1 are complicated by the fact that i) the soot concentration decreased during the PM sampling duration and ii) the DustTrak failed to measure correctly. However, we think that there is merit in presenting the experiments in the same order in which they were performed. In this way, the readers can follow the progress achieved in the course of the experiments: in the case of Aerosol 2 and 3 both technical issues were rectified. If we were to rearrange the order in which the experiments are presented and discussed, the readers would be left with the impression that the aerosol generation cannot be stabilised, which is not true. The experiments with Aerosols 2 and 3 (as well as subsequent experiments we have performed since then) prove that the aerosol generation can remain stable over several hours.

If the Reviewer agrees, we would suggest to leave the discussion (and Figure 6) as is.

Reviewer 2: In general the composition of the three model aerosols seems similar and it's unclear if we are meant to focus on any differences? The main differences seem to be RH and temperature.

Answer: Model aerosols 1,2 and 3 differ in temperature, RH, mass concentration and chemical composition. It is true that the difference in chemical composition is not so pronounced: The % mass fractions of the different chemical constitutents was varied in the range $\approx 30-40$ % OM, $\approx 5-15$ % EC, $\approx 7-15$ % nitrate, $\approx 5-15$ % sulphate and $\approx 10-20$ % mineral dust. This is in accordance with past studies on European continental aerosols. See, for instance, the pie charts with the chemical composition of urban, suburban and rural PM10 aerosols in Switzerland (page 7, in german):

https://www.empa.ch/documents/56101/246436/chem_char_pm10/d7e07ec3-0442-4749-b34f-3d26c9687038

We could easily increase the % mass fraction of any of the chemical components (OC, OM, inorganic ions, mineral dust) up to at least 80 %. However, such aerosols would not be representative of typical ambient aerosols any more. To summarise: By generating model aerosols 1, 2 and 3 we wanted to showcase the potential of the facility to produce "customised" aerosols which differ in T, RH, mass concentration and chemical composition. The focus of the study was on European continental aerosols.

Reviewer 2: Line 355: The role of hygroscopicity and aerosol liquid water in influencing the performance of particle mass sensors (e.g. Crilley et al., 2018; Di Antonio et al, 2018) should be mentioned. The evaporation of water in the TEOM due to heating may be at least as important as nitrate/organic matter. Liquid water is a potential source of difference for all the online methods (70% RH) when comparing to the gravimetric reference (47.5% RH) especially if hygroscopic components effloresce on the filter. In future, the water content of the aerosol could be estimated from composition, RH and temperature using a thermodynamic model such as E-AIM (http://www.aim.env.uea.ac.uk/aim/aim.php).

Answer: Thank you to the Reviewer for the valuable comments. We have modified the sentence referring to TEOM artefacts as follows: "Since the aerosol stream sampled by the TEOM 1405 is heated to 30 °C, a fraction of the (semi)volatile components (e.g. nitrate, secondary organic aerosol and water) evolves into the gas phase and is therefore not collected on the filter". We have also modified the following section on light-scattering PM monitors: "Deviations may occur if the built-in functions differ substantially from the real density function of the aerosol. Hygroscopic growth of aerosol particles can also lead to considerable measurement artefacts especially when low-cost PM sensors are used (Di Antonio et al., 2018; Crilley et al., 2018))". In the future, we will try to estimate the water content by using a thermodynamic model such as E-AIM (http://www.aim.env.uea.ac.uk/aim/aim.php).

Reviewer 2: Line 369: Capitalise "in".

Answer: Thank you to the Reviewer for spotting this typo. The sentence now begins with a capital letter "In...".

Reviewer 2: Line 390: The data in the paper focuses on 20-40 ug/m3. It's hard to conclude from this that the stable operating range is "few to 500 ug/m3". Please include some additional evidence or modify this statement.

Answer: The Reviewer is right that the experiments presented in the manuscript were performed in the mass concentration range 20 - 40 $\mu$g/m3. The reason for this is that the EU target values (annual mean values) for PM2.5 and PM10 are 25 $\mu$g/m3 and 40 $\mu$g/m3, respectively, and we wanted to focus on mass concentration ranges which are relevant for the EU member states. We can, however, generate up to about 80 $\mu$g/m3 of fresh soot (depending on particle size), 150 $\mu$g/m3 of aged soot, >500 $\mu$g/m3 of inorganic salts and >500 $\mu$g/m3 of mineral dust in the flow tube homogeniser (i.e. after dilution). Below please find a figure that presents PM10 mass concentration measured by a low-cost optical sensor. The PM sensor reports an average concentration of about 650 $\mu$g/m3. The reference concentration was 925$\pm$60 $\mu$g/m3. This measurement shows that the facility can also produce higher particle loadings in a stable manner.

Reviewer 2: Figure 6: Please use different marker shapes as well as colours for the different instruments.

Answer: We thank the Reviewer for this suggestion. We have modified Fig. 6 so that each data set is marked with a different symbol (circles for TEOM, squares for the DustTrak and triangles for the Fidas Frog). The datasets were already marked in different colours but they might be hard to distinguish if the manuscript is printed on grayscale.

[Figure]

**Fig. 1.** Calibration of low-cost optical sensor at PM10 mass concentrations larger than 500 um/m3.